# Whole-exome sequencing identifies protein-coding variants associated with brain iron in 29,828 individuals

Weikang Gong [1,2,8] ✉, Yan Fu [3,8], Bang-Sheng Wu [1,8], Jingnan Du [4,8], Liu Yang [1], Ya-Ru Zhang [1], Shi-Dong Chen [1], JuJiao Kang [5,6], Ying Mao [1], Qiang Dong [1], Lan Tan [3], Jianfeng Feng [5,6,7], Wei Cheng [1,5,6] ✉ & Jin-Tai Yu [1] ✉

Iron plays a fundamental role in multiple brain disorders. However, the genetic underpinnings of brain iron and its implications for these disorders are still lacking. Here, we conduct an exome-wide association analysis of brain iron, measured by quantitative susceptibility mapping technique, across 26 brain regions among 26,789 UK Biobank participants. We find 36 genes linked to brain iron, with 29 not being previously reported, and 16 of them can be replicated in an independent dataset with 3,039 subjects. Many of these genes are involved in iron transport and homeostasis, such as *FTH1* and *MLX*. Several genes, while not previously connected to brain iron, are associated with iron-related brain disorders like Parkinson's (*STAB1, KCNA10*), Alzheimer's (*SHANK1*), and depression (*GFAP*). Mendelian randomization analysis reveals six causal relationships from regional brain iron to brain disorders, such as from the hippocampus to depression and from the substantia nigra to Parkinson's. These insights advance our understanding of the genetic architecture of brain iron and offer potential therapeutic targets for brain disorders.

Iron is crucial for numerous physiological processes, including neurotransmitter synthesis, myelin formation, DNA synthesis and mitochondrial functions, and it profoundly influences neurodevelopment, cognition, and brain outcomes[1,2]. Our brain maintains a precise regulation of iron homeostasis. Any disturbance in this intricate balance, whether through iron overload or deficiency, may result in the emergence of brain disorders[3]. For example, iron accumulation might facilitate neuronal cell death in some neurodegenerative diseases, such as Alzheimer's disease (AD)[4]. In addition, the substantia nigra often has excess iron in Parkinson's disease (PD), which possibly promotes oxidative stress and neuronal damage[5]. Moreover, cerebral iron deficiency, linked to alterations in hippocampal glucocorticoid receptor signaling, has been implicated in inducing depression[6]. Given iron's important role in brain development and its connection to multiple brain disorders, understanding the genetic architecture of brain iron accumulation can provide insights into brain development and the underlying mechanisms of these disorders, allowing for designing better diagnostic and therapeutic strategies.

Quantitative susceptibility mapping (QSM) is an emerging technique that enables the non-invasive measurement of brain iron

[1]School of Data Science, Department of Neurology and National Center for Neurological Disorders, Huashan Hospital, State Key Laboratory of Medical Neurobiology and MOE Frontiers Center for Brain Science, Fudan University, Shanghai, China. [2]Centre for Functional MRI of the Brain (FMRIB), Nuffield Department of Clinical Neurosciences, Wellcome Centre for Integrative Neuroimaging, University of Oxford, Oxford OX3 9DU, UK. [3]Department of Neurology, Qingdao Municipal Hospital, Qingdao University, 266071 Qingdao, China. [4]Department of Psychology, Center for Brain Science, Harvard University, Cambridge, MA 02138, USA. [5]Institute of Science and Technology for Brain-Inspired Intelligence, Fudan University, 200433 Shanghai, China. [6]Key Laboratory of Computational Neuroscience and Brain-Inspired Intelligence, Ministry of Education, Fudan University, 200433 Shanghai, China. [7]Department of Computer Science, University of Warwick, Coventry, UK. [8]These authors contributed equally: Weikang Gong, Yan Fu, Bang-Sheng Wu, Jingnan Du. ✉e-mail: weikanggong@fudan.edu.cn; wcheng@fudan.edu.cn; jintai_yu@fudan.edu.cn

levels with high spatial resolution and sensitivity[7]. Built upon susceptibility-weighted MRI (swMRI), QSM has been demonstrated to be more sensitive to reflect tissue iron contents both phenotypically and genetically than other swMRI-derived measures, such as T2*, and has shown higher robustness to acquisition noises and increased reproducibility[8]. A high positive correlation between brain iron level and QSM has been established from postmortem studies[9]. Previous studies also showed that brain iron has a high heritability in multiple regions, such as the putamen, substantia nigra, and pallidum[8]. While genome-wide association studies (GWAS) have found several loci associated with brain iron, these findings remain constrained to a few brain regions and common genetic variations (minor allele frequency > 1%)[8]. Additionally, many loci identified by GWAS map to noncoding regions of the genome, posing challenges in exploring the underlying mechanism. To overcome these limitations, a powerful technique, whole-exome sequencing (WES)[10], can be used to identify protein-coding variants that are associated with brain iron. A large-scale exome-wide association analysis on multiple brain regions can uncover the intricate genetic architecture of brain iron accumulation and potentially highlight neural pathways crucial to iron-related brain disorders.

In this study, we conducted the most extensive exome-wide association study (EWAS) of brain iron accumulation to date. Leveraging genetic, brain imaging and phenotypic data from 26,789 subjects in the UK Biobank dataset, we systemically identified protein-coding variants associated with brain iron and studied the relationships between iron-related genes and brain disorders and phenotypes. Specifically, this study has four major goals. Firstly, we will identify rare and common genes that are associated with brain iron accumulation across multiple brain regions covering subcortical and cerebellar structures. Secondly, we aim to explore the biological functions of the identified genes, such as the biological pathways in which they are enriched in. Thirdly, we aim to explore the relationships between brain iron-related genes and disorders, including whether regional brain iron accumulation has causal relationships to multiple brain disorders. Finally, we used phenome-wide association study (PheWAS) to identify genetic associations of brain iron-related genes with a broad set of phenotypic variables.

## Results

### An overview of data and analysis pipeline
Our study primarily used brain imaging and phenotypic and genetic data from the UK Biobank, including 26 regional QSM features extracted from the swMRI data, exome-sequencing data, and diverse phenotypes for phenome-wide association studies. In the primary analysis of EWAS, we included a total of 29,828 individuals of white British ethnicity without illness conditions of brain cancer, stroke, or dementia, aged between 40 and 69, with ~52% of them being females. Among them, 26,789 of them are in the discovery set and the remaining 3039 are in the replication set (see the "Methods" section). A summary of the demographic information is provided in Table 1. A total of 18,800 rare genes and 41,790 common variants were analyzed in this study (see the "Methods" section).

A comprehensive visualization of our analysis pipeline is provided in Fig. 1. The study was initiated with the discovery of EWAS (N = 26,789) to identify genetic variants correlated with brain iron as measured by QSM techniques. (Fig. 1a and b). Validation and replication studies (N = 3039) were conducted to verify the robustness of our findings (Fig. 1c). Further, we examined the functionality and organization of the identified genes (Fig. 1d) and performed Mendelian randomization (MR) analysis to investigate the potential causal relationship between brain iron and multiple brain disorders (Fig. 1e). Lastly, a PheWAS was conducted to explore a broad set of brain iron-phenotype associations (Fig. 1e).

**Table 1 | Demographic information of participants in this study**

|  | Discovery set | Replication set |
|---|---|---|
| *N* | 26,789 | 3039 |
| Age (years; mean ± sd) | 55.22 (7.43) | 52.92(7.34) |
| Sex (female; percent) | 14,041 (52.4) | 1621 (53.3) |
| BMI | 26.57 (4.17) | 26.34 (4.14) |
| **Educational qualification (%)** | | |
| A levels | 1520 (5.7) | 172 (5.7) |
| College | 12,061 (45.2) | 1404 (46.3) |
| CSE | 726 (2.7) | 88 (2.9) |
| None of above | 1830 (6.9) | 138 (4.6) |
| NVQ | 4142 (15.5) | 529 (17.5) |
| O levels | 2966 (11.1) | 332 (11.0) |
| Other | 3451 (12.9) | 368 (12.1) |
| **Smoking status (%)** | | |
| Never | 16,335 (61.1) | 1938 (63.9) |
| Previous | 8863 (33.1) | 932 (30.7) |
| Current | 1543 (5.8) | 164 (5.4) |
| **Drinking status (%)** | | |
| Never | 519 (1.9) | 58 (1.9) |
| Previous | 552 (2.1) | 46 (1.5) |
| Current | 25,712 (96.0) | 2935 (96.6) |

### Rare protein-coding variants associated with brain iron
We conducted an exome-wide association study on 26,789 subjects to identify rare variants associated with brain iron levels across 26 brain regions. Out of the resulting associations, a total of 207 reached exome-wide significance (Bonferroni corrected $p < 1.7 \times 10^{-8}$, as detailed in the "Methods" section), covering 24 out of the 26 investigated brain regions (no associations were found for the left and right amygdala) (Figs. 2a and 3, Supplementary Data 1). These identified variants were mapped to 20 different protein-coding genes, with 18 of them not being previously reported in GWAS, while 2 overlapped with findings from a previous study[8]. The Manhattan plots and Q–Q plots for each brain region can be found in Supplementary Fig. S1. We further performed a replication study using 3039 subjects (see the "Methods" section). From the previously mentioned 20 genes, 4 are significant in both sets ($p < 0.05$), encompassing 24 gene-based associations (Supplementary Data 2). This was notable considering the number of genes to replicate is just one under the null hypothesis of no associations.

We performed a power analysis of our WES analysis based on the method of a previous study[11]. The range of variants' effect sizes is from 0.03 to 0.08 based on an analysis of our WES results (the regression coefficients and standard error)[11]. Therefore, given a sample size of $N = 29,828$, the estimated power is from 39.7% to 100% for the smallest effect to the largest to reach the significance level of uncorrected $p < 1.7 \times 10^{-8}$ (Supplementary Data 3).

Furthermore, leave-one-variants out (LOVO) analysis was conducted for each of the significant associations to assess the sensitivity of our findings to analytical approaches (see the "Methods" section). The results showed that over 99.4% of the significant associations involving the above 20 significant genes remained significant in the LOVO analysis. This suggests that most significant associations arise from multiple contributing rare variants. There were a few exceptions. For instance, the association between the thalamus and *ULBP2* was influenced by single variants. The LOVO findings are cataloged in Supplementary Data 4. In addition, we conducted a conditional analysis to evaluate whether the identified rare variant signals were independent of nearby common variants (see the "Methods" section). As a result, all variants are significant, indicating that the rare variants

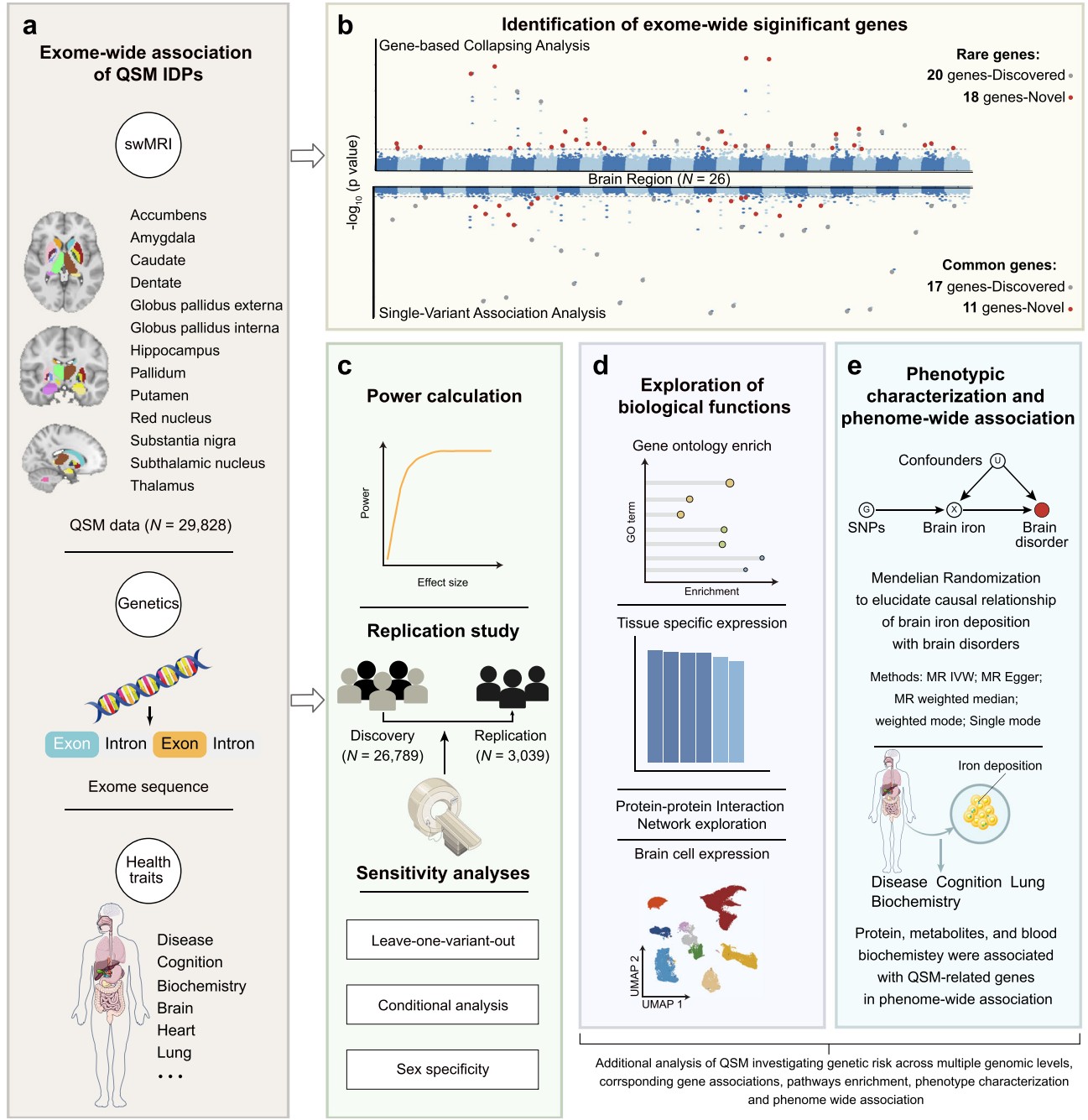

**Fig. 1 | A summary of the analysis pipeline of the current study. a** The UK Biobank data used in the current study, including quantitative susceptibility mapping (QSM) features derived from susceptibility-weighted MRI, whole-exome sequencing data and phenotypic data. **b** Exome-wide association study (EWAS) of brain iron with rare and common variants. **c** Validation study and replication analysis of EWAS. **d** Post EWAS analysis, including gene set enrichment analysis, protein–protein interaction, tissue and single-cell expression, and **e** phenotype-wide association study and Mendelian randomization analysis. swMRI susceptibility-weighted MRI, QSM quantitative susceptibility mapping, MR Mendelian randomization, IVW inverse variance weighted. This figure was partly generated using Servier Medical Art, provided by Servier, licensed under a Creative Commons Attribution 4.0 unported license.

signals are independent of the nearby common variants (Supplementary Data 5). Additionally, sex-specific EWAS were executed. We found that 12 of the 20 rare genes are significant for males, and for females, 9 of the 20 rare genes are significant ($p < 1.7 \times 10^{-8}$). (Supplementary Data 6).

**Common protein-coding variants associated with brain iron**
Expanding upon the above analyses, we performed an EWAS to identify common protein-coding variants associated with brain iron (see the "Methods" section). In summary, 105 associations were discovered

that pass the genome-wide significance threshold (Bonferroni-corrected $p < 4.6 \times 10^{-8}$, as detailed in the "Methods" section) (Figs. 2b, 3, Supplementary Data 7). The identified variants map to 17 genes, with 11 of them not being previously reported[8]. It is worth noting that, as a comparison to ref. 8, our analyses specifically focused on variants on the exome regions and used a broader array of brain regions of interest. Manhattan plots and Q–Q plots for each brain region are shown in Supplementary Fig. S2.

As in the rare variants analysis, we performed a replication study. Among the above 17 identified genes, 13 of them can still be found

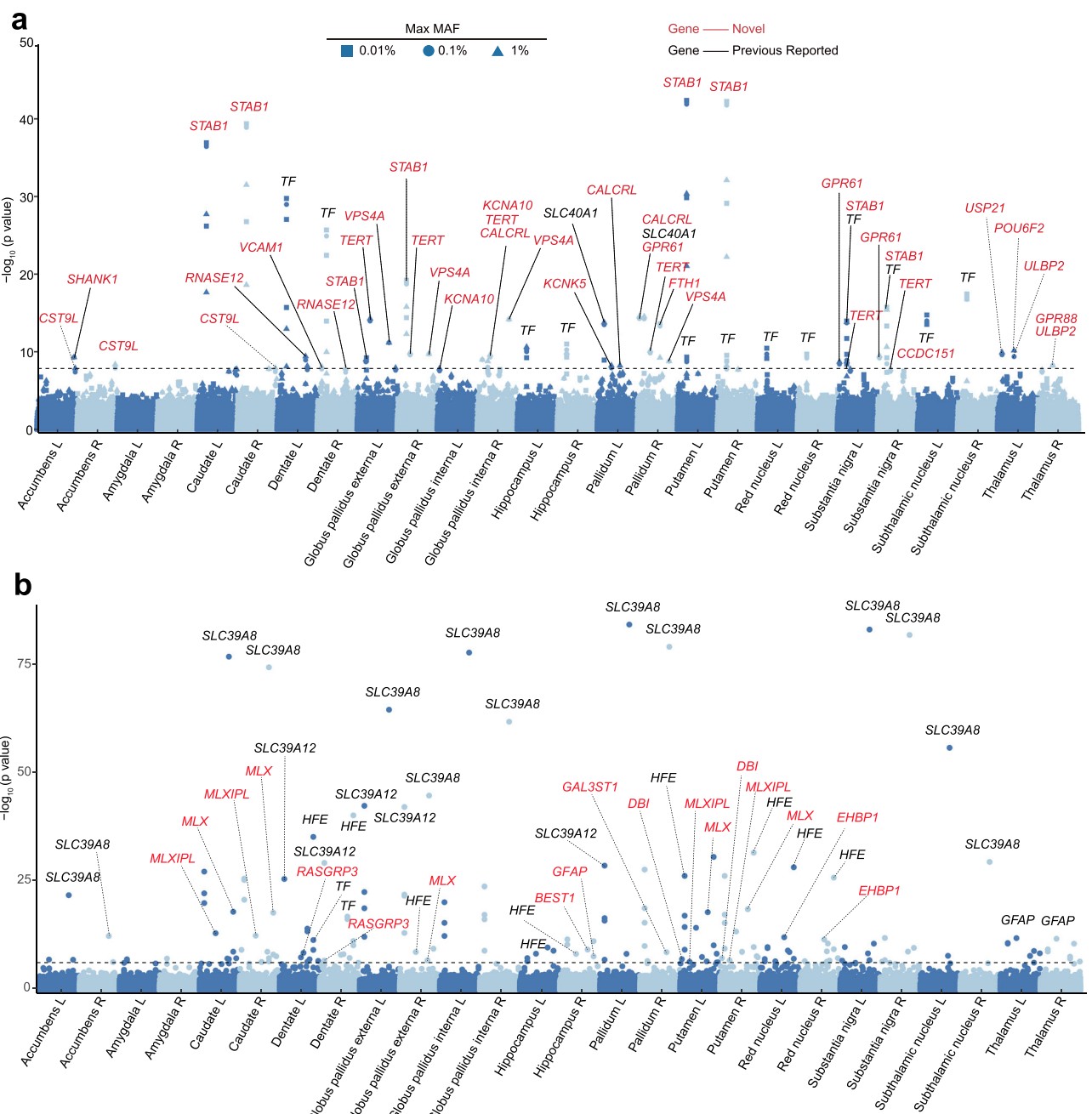

**Fig. 2 | Exome-wide association analysis of rare and common protein-coding genes with brain iron across 26 regions. a** Significance levels of genes that are mapped from rare variants. **b** Significance levels of genes that are mapped from common variants. The *p*-values reported are two-sided and unadjusted. The *x*-axis represents the brain regions analyzed in the current study (L and R represent left and right brain regions), and the *y*-axis represents the $-\log_{10}$ *p*-values of each gene. The gray dashed line is the exome-wide significance threshold, based on the Bonferroni-corrected $p < 0.05$. Genes in red are discovered by the current study, and in black are previously reported. MAF minor allele frequency, R right, L left.

significant (Supplementary Data 8, uncorrected $p < 0.05$). In expectation, the number of genes that can be replicated is one.

## Brain iron-associated genes are significant in existing GWAS of brain-related traits and diseases

We performed a literature search to explore how the 36 brain iron-associated genes in our analysis overlap with existing GWAS findings. We found that 3 genes are significant in AD GWAS, 4 genes are associated with cognitive-related traits, 2 genes are associated with depression-related traits, 1 gene is associated with PD, and 7 genes are reported in bipolar disorder and schizophrenia GWAS. The full list of our literature search results is shown in Supplementary Data 9.

## Functional enrichment and biological validation of brain iron-associated genes

To dive deeper into and verify the biological attributes of the identified genes, we conducted a functional enrichment analysis. Results demonstrated that brain iron-related genes are robustly enriched in iron-related functions. Notably, the pathway of intracellular iron ion homeostasis showed the highest statistical significance ($p = 3.5 \times 10^{-12}$) followed by iron ion homeostasis ($p = 3.4 \times 10^{-11}$) and transition metal ion transport ($p = 7.4 \times 10^{-11}$) (Fig. 4a, Supplementary Data 10). Furthermore, the protein−protein map of these genes forms a dense network associated with iron, including key genes in the *SLC* families and *FTH1* (visible as brown clusters in Fig.4b, Supplementary Data 11).

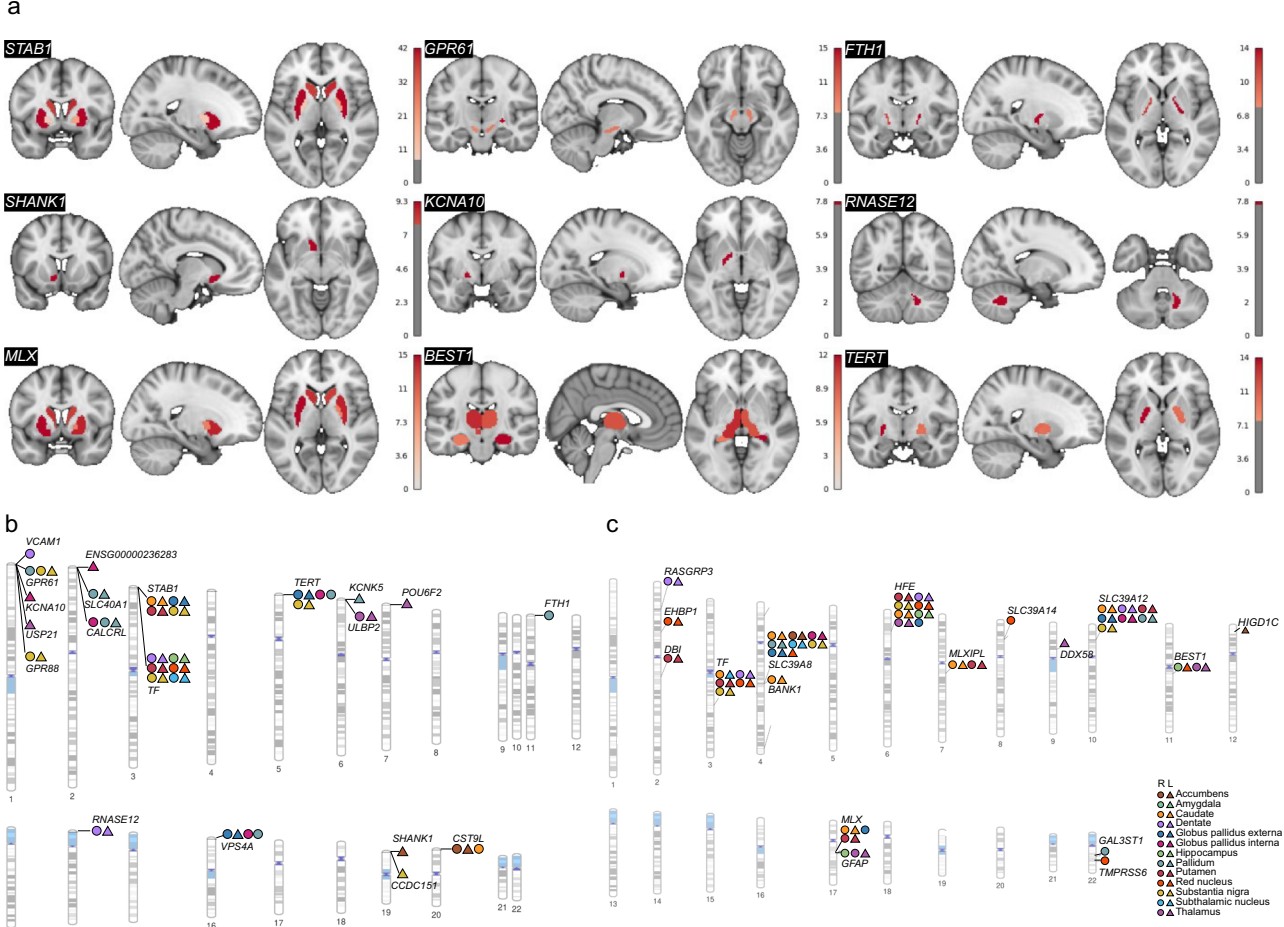

**Fig. 3 | Brain-wide and chromosome-wide mapping of brain iron-associated genes. a** Brain-wide association map of 16 example genes that are associated with brain iron. **b** Ideogram of rare genes that influence brain iron. **c** Ideogram of common genes that influence brain iron. R right, L left.

We then tested whether the identified genes were differentially expressed across various tissues, using the GTEx database (12). Notably, our genes exhibited significant differential expression in brain tissues compared to others (Fig. 4c). Notably, these included regions in our analysis, such as the substantia nigra, putamen, and cerebellum ($p<1.0\times10^{-4}$) (Fig. 4c, Supplementary Data 12). Leveraging single-cell RNA sequencing data of the human brain, we found that brain iron-associated genes showed higher expression levels in both excitatory and inhibitory neurons (Fig. 4d). Removing the top associated genes still kept most of the enriched pathways and differential expressed genes significantly (Supplementary Fig. S3, Supplementary Data 13 and 14). These analyses verify the biological relevance of our findings across both rare and common protein-coding variants.

## Mendelian randomization analysis of regional brain iron and brain disorders

Brain iron plays a crucial role in various brain disorders. To investigate whether regional brain iron has causal relationships with brain disorders, we conducted a two-sample Mendelian randomization (MR) analysis. Four brain disorders were selected for this study: depression, bipolar disorder, PD, and AD. Our primary focus was to understand the influence of brain iron on these disorders. Therefore, regional brain iron features were used as exposures, while the brain disorders were used as outcomes in the MR analysis (see the "Methods" section).

Our investigation revealed 11 significant causal relationships for the four candidate brain disorders (Table 2, Supplementary Data 15 and 16): (1) from the subthalamic nucleus, accumbens and

thalamus to bipolar disorder (top $p=3.6\times10^{-7}$, FDR $=2.2\times10^{-5}$); (2) from the caudate, substantia nigra, dentate and putamen to PD (top $p=1.0\times10^{-4}$, FDR $=0.0028$); (3) from the hippocampus and subthalamic nucleus to depression (top $p=7.6\times10^{-3}$, FDR $=0.046$). Additionally, we also conducted a reverse MR analysis (from disease to regional brain iron), revealing that the above significant associations are not significant in this analysis (Supplementary Data 17), indicating that the direction of causality was not biased by reverse causation. It is worth noting that while previous studies have documented statistical associations[12–14], our results uncovered potential causal linkages from the brain to diseases.

Several sensitivity analyses were also conducted to assess the robustness of our results. MR results from alternative approaches are shown in Supplementary Data 16 and 18. The MR-Egger intercept was close to zero, and the pleiotropy test was not significant (Supplementary Data 15 and 17), suggesting that there was no directional pleiotropy in our analysis. In addition, we performed multivariate MR to assess the independence of causal effects (see the "Methods" section). The results revealed that the causal association between iron levels in 26 brain regions and brain disorders was not significant (Supplementary Data 19). This observation suggests that the causal relationship between brain iron levels and brain disorders may not be entirely independent across different brain regions.

## PheWAS of brain iron-associated genes

Brain iron plays a pivotal role in numerous fundamental biological processes. Conducting a PheWAS enables us to attain a comprehensive understanding of the diverse impacts of brain iron across the

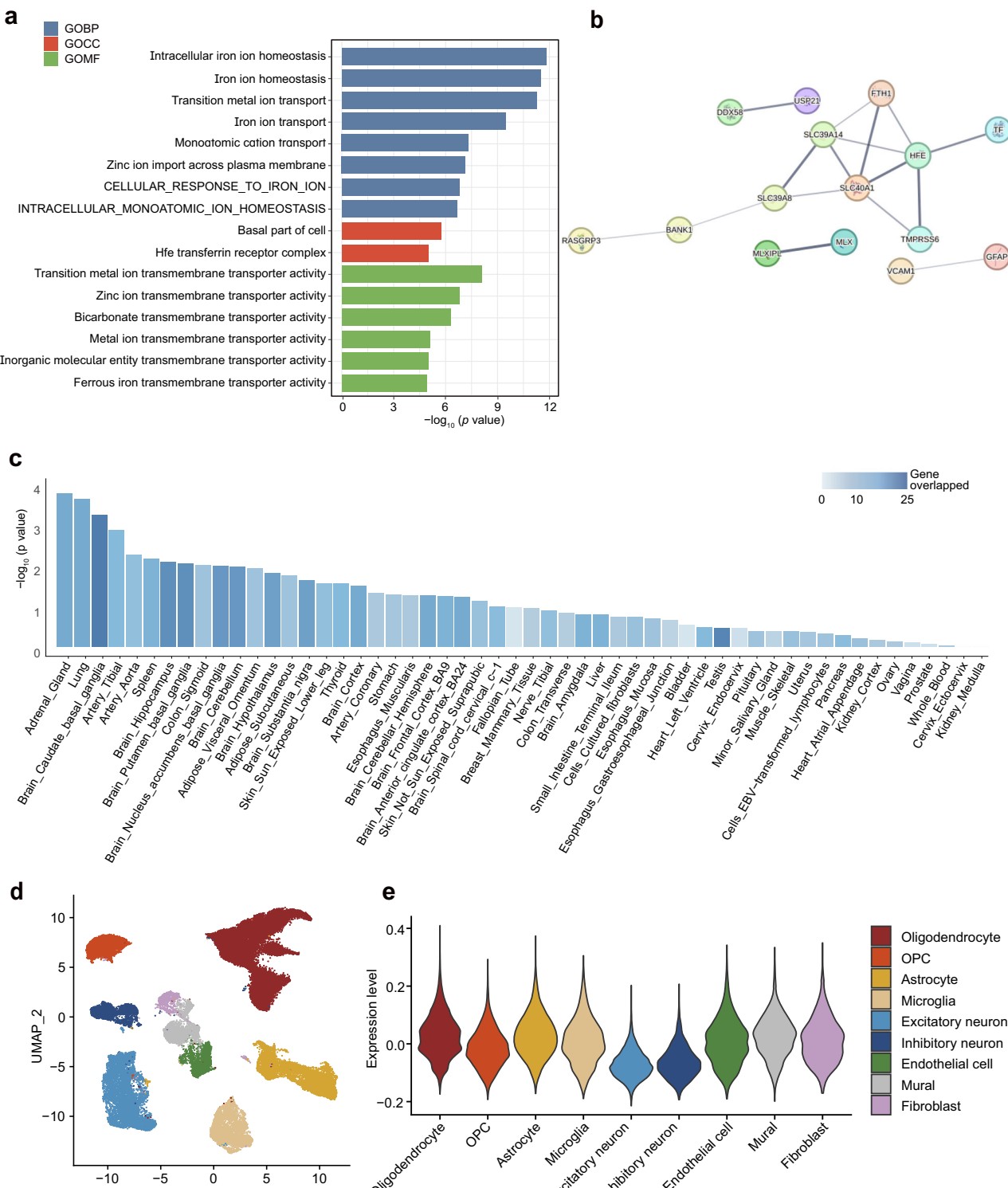

**Fig. 4 | Biological functions of genes associated with brain iron. a** Gene set enrichment analysis of significant genes identified in the exome-wide association analysis, using Gene Ontology and KEGG Ontology database. The *p*-values reported are two-sided and unadjusted. **b** Protein–protein interaction network of significant brain iron-related genes. **c** Tissue-wide differential expression analysis of significant brain iron-related genes using GTEx dataset. The *p*-values reported are two-sided and unadjusted. **d** Uniform manifold approximation and projection (UMAP) visualization of human brain single-cell sequencing data. **e** Expression levels of significant brain iron-related genes in human brain single-cell sequencing data. The colors indicate the tissue types. GOBP gene ontology biological process, GOCC gene ontology cellular component, GOMF gene ontology molecular function, OPC oligodendrocyte progenitor cell.

phenotypic landscape and enhances the validity of the findings of the current study[15–17].

The phenotypes considered in this analysis encompass cognition metrics, neurological and psychiatric conditions, blood chemistry measures, neuroimaging phenotypes, and plasma protein levels. Among the assessed phenotypes, the most robust associations were observed between brain iron-related genes and plasma proteins. *STAB1* is associated with multiple plasma protein, including, e.g. MME

**Table 2 | Causal relationships identified by Mendelian randomization analysis**

| Brain iron (exposure) | Diseases (outcome) | Beta | p value | FDR |
|---|---|---|---|---|
| Subthalamic nucleus (right) | Bipolar | 0.011 | 3.59E−07 | 2.15E−05 |
| Accumbens (left) | Bipolar | 0.023 | 3.11E−05 | 9.33E−04 |
| Substantia nigra (right) | PD | 0.010 | 1.45E−04 | 2.90E−03 |
| Substantia nigra (left) | PD | 0.011 | 1.33E−03 | 1.92E−02 |
| Caudate (right) | PD | 0.013 | 1.60E−03 | 1.92E−02 |
| Dentate (right) | PD | 0.006 | 4.25E−03 | 4.25E−02 |
| Putamen (left) | PD | 0.010 | 5.11E−03 | 4.34E−02 |
| Thalamus (left) | Bipolar | 0.024 | 5.79E−03 | 4.34E−02 |
| Putamen (right) | PD | 0.010 | 6.71E−03 | 4.47E−02 |
| Subthalamic nucleus (right) | Depression | 0.003 | 7.67E−03 | 4.60E−02 |
| Hippocampus (right) | Depression | 0.017 | 8.49E−03 | 4.63E−02 |

The iron in brain regions are the exposure variables and the brain disorders are the outcome variables.

$(p = 6.7 \times 10^{-13})$, ECE1 $(p = 4.0 \times 10^{-9})$, OMD $(p = 9.2 \times 10^{-8})$, LTBP2 $(p = 5.6 \times 10^{-6})$. *TERT* is associated with proteins EDA2R $(p = 1.4 \times 10^{-6})$ and FOLR3 $(p = 5.7 \times 10^{-6})$. *SHANK1* is associated with protein NTRK2 $(p = 8.9 \times 10^{-6})$ (Fig. 5a). Genes mapped to common variants have more significant levels of associations, as shown in Fig. 5b.

Our exploration of cognitive metrics and disease phenotypes revealed five significant gene-phenotype associations, all mapped to common variants (FDR-corrected $p < 0.05$): *SLC39A8* is associated with fluid intelligence $(p = 1.4 \times 10^{-10})$ and prospective memory $(p = 1.5 \times 10^{-9})$; *HFE* is associated with multiple sclerosis $(p = 2.2 \times 10^{-6})$, fibrosis liver $(p = 5.3 \times 10^{-9})$ and broader liver disease $(p = 6.5 \times 10^{-5})$; *EHBP1* is associated with the fluid intelligence $(p = 9.7 \times 10^{-5})$; *MLX* is associated with migraine $(p = 1.2 \times 10^{-3})$ (Fig. 5b). These associations were especially noteworthy as these genes ranked among the most significantly associated in our primary association analyses (Fig. 2b).

In other phenotypic variables that we analyzed, for genes mapped to rare variants, *RNASE12* is associated with the insulin-like growth factor 1 (IGF-1, $p = 1.0 \times 10^{-4}$). Common variants show stronger phenotype-wide associations (Fig. 5b). For instance, the gene *BEST1* is associated with the ratio of omega-6 to omega-3 fatty acids $(p = 1.4 \times 10^{-38})$, as determined by nuclear magnetic resonance (NMR) spectroscopy.

## Discussion

In our large-scale EWAS of brain iron, we uncovered the genetic underpinnings of brain iron accumulation using high-quality genetic data and QSM from swMRI scans of 29,828 UK Biobank participants. Our discovery of EWAS revealed 36 genes ($N = 26,789$), mapped from either rare or common protein-coding variants, associated with brain iron accumulation. Remarkably, 29 of these genes were not previously reported. 16 of them can be replicated in an independent dataset with 3039 individuals. Our functional enrichment analysis revealed that the identified genes are enriched in biological pathways involving ion transport and homeostasis. MR analysis identified several interesting causal relationships between regional brain iron accumulation to brain disorders, such as from the substantia nigra to PD, and from the accumbens nucleus to AD, and from the hippocampus to depression. Furthermore, our phenotype-wide association study highlighted genes like *SLC39A8* and *EHBP1*'s associations with fluid intelligence and *HFEs* linked to multiple sclerosis and liver-related diseases.

Our analyses revealed multiple rare genes that have been previously proposed as important contributors to the development of brain disorders or related to iron transport and ferritin (Fig. 3a, b). For example, the gene *STAB1* has the overall highest significant level in our EWAS and displayed replicated associations with multiple brain regions, including those of the putamen, caudate, globus pallidus externa and substantia nigra. It has previously been reported as a candidate gene for bipolar disorder[18] and cerebrovascular diseases[19]. *FTH1* is associated with the pallidum in our EWAS. This gene was closely linked to various brain disorders, as it is responsible for encoding the heavy chain of ferritin[20]. Our rare variants analysis also identified the transferrin gene (*TF*) that is associated with the hippocampus. This gene was known to participate in iron transport. It exhibited an interesting convergence of rare and common variant evidence. *KCNA10* is associated with globus pallidus interna, whose changes in expression level were observed in PD patients[21].

For common genes, owing to the involvement of a larger number of candidate brain regions than the previous study[8], we identified several gene–brain associations(Fig. 3a, c). For example, *HFE* was associated with the dentate nucleus and red nucleus, which has been previously reported to lead to an increased risk of developing movement disorders[22]. The gene *MLX* is associated with putamen. This gene controls the transport and storage of ferrous iron[23].

Importantly, our EWAS identified eight genes linked to the brain iron levels of substantia nigra (SN): *STAB1*, *TF*, *GPR61*, *TERT*, *SLC39A8*, *SLC39A12* and *HFE*. These genes are linked to PD's underlying mechanisms. For instance, previous research has suggested that dysregulated STAB1 expression in microglia might play a role in the pathogenesis of PD[24]. The gene *SLC39A8*, encoding a metal ion transporter, has been linked to various conditions, including PD[25]. Interestingly, our MR analyses further revealed significant causal relationships between brain iron accumulation in the SN to PD. PD is marked by motor impairments, stemming largely from the loss of dopamine-producing neurons in the SN[26]. Dopamine, a pivotal neurotransmitter, governs our motor functions and coordination. A deficit in dopamine results in multiple movement challenges in PD patients, such as tremors, stiffness, bradykinesia, and balance issues. Notably, increased iron accumulation within the substantia nigra has been observed in PD patients, complementing our MR results and indicating a potential causal link between the SN and PD. Diving deeper into these biological pathways could enrich our understanding of PD, potentially leading to more targeted and effective therapeutic interventions[27].

We identified four genes associated with the hippocampal iron level: *BEST1*, *HFE*, *TF*, and *GFAP*. These genes may provide insights into the connection between the hippocampus and depression. For example, Astrocytes, characterized by their expression of *GFAP*, play an important role in the central nervous system. They are abundant in the hippocampus, a central component of the limbic system that has long been theorized to play a role in depression's neuropathology[28]. Post-mortem studies have highlighted the involvement of cerebral astrocytes immunoreactive to *GFAP* in the pathogenesis of depression[29,30]. In addition, the *HFE* gene is primarily known for its role in hereditary hemochromatosis, a genetic disorder that causes the body to absorb too much iron. Mutations in the *HFE* gene can lead to excessive iron accumulation. Notably, the brain's uptake of blood iron is important for the optimal synthesis of neurotransmitters such as serotonin, dopamine, and noradrenaline. These neurotransmitters, involved in regulating emotional behaviors, rely on neuron aromatic hydroxylase, with iron acting as an important cofactor. Notably, noradrenaline affects neuroplasticity through the brain-derived neurotrophic factor, which is important for the functioning of prefrontal and hippocampal neurons implicated in depression[31]. Moreover, research has also established a correlation between severe depression symptoms and high body iron levels[32,33] as well as increased brain iron levels[14]. Our MR analysis revealed a significant causal linkage between the hippocampus and depression. In studies using mouse models, cerebral iron deficiency, leading to the suppression of the hippocampal glucocorticoid receptor signaling pathway, has been implicated in inducing depression[6]. A consistent observation among

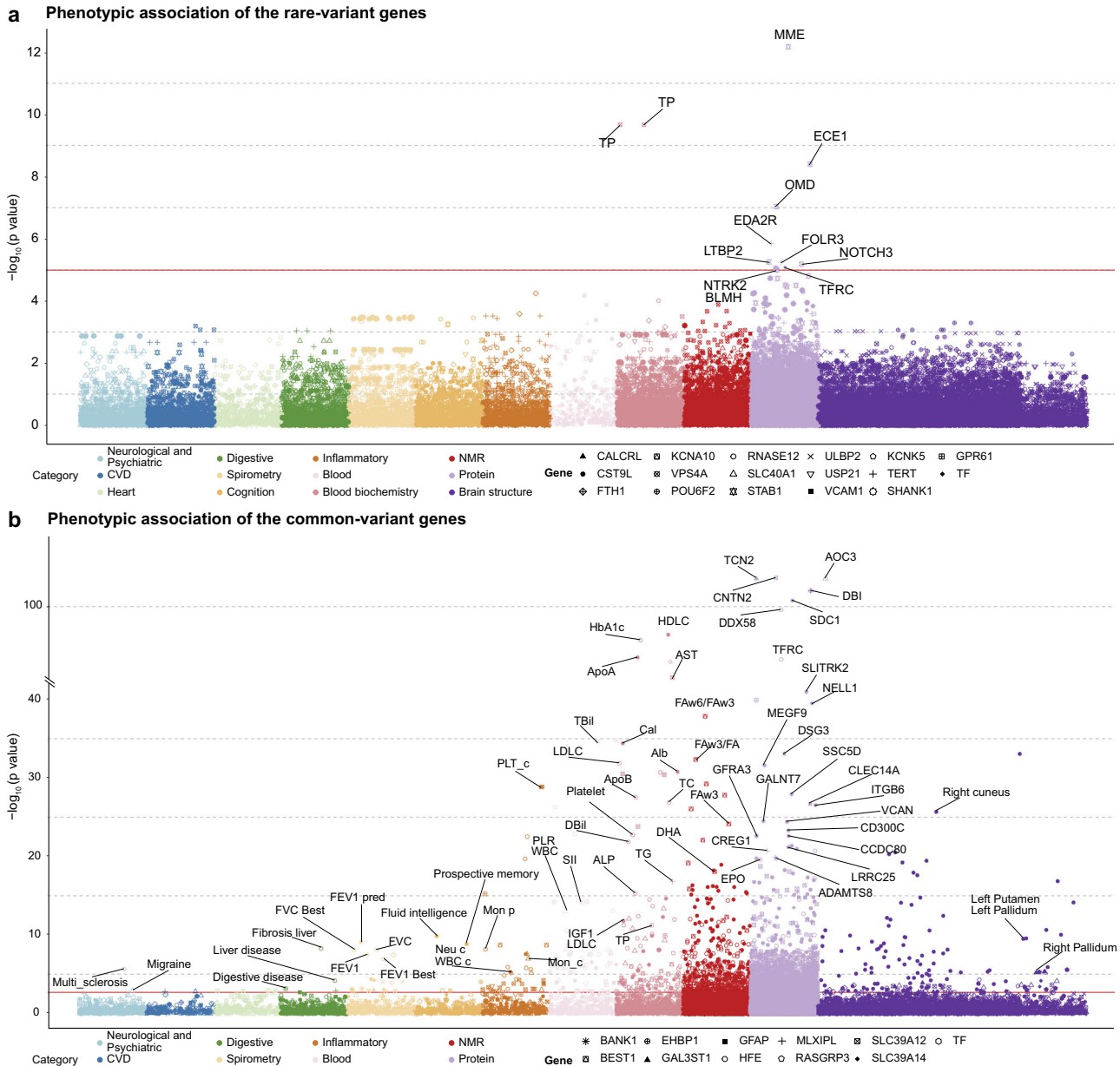

**Fig. 5 | Phenome-wide association analysis of brain iron-associated genes.**
**a** Phenome-wide associations of genes mapped from rare variants. **b** Phenome-wide associations of genes mapped from common variants. Scatterplot showing associations between brain iron-related genes and a wide range of phenotypes, including 12 categories, listed at the bottom left of each figure. The *y*-axis indicates the −log$_{10}$ of the *p*-value for each association, and the *x*-axis represents different phenotype categories. The *p*-values shown are two-sided and unadjusted for multiple testing. Linear regression models and SKAT-O tests were used for gene-based

analysis, and the model adjusted for age, gender, and top 10 ancestral principal components. Red line in each figure is the FDR 0.05 correction threshold. CVD cardiovascular disease, FEV1 forced expiratory volume in 1 s, FEV1_Best best measure of FEV1, FEV1_predperc predicted percentage of FEV1TP, total protein, HDLC high-density lipoprotein cholesterol, LDLC low-density lipoprotein cholesterol, WBC white blood cell count, Neu_c neutrophil count, PLT platelet count, HBA1C glycated hemoglobin, PLR platelet-to-lymphocyte ratio, SII systemic immune-inflammation index.

depressed individuals is reduced hippocampal volume, correlating with the length and recurrence of depressive episodes[34,35]. Furthermore, the hippocampus closely interacts with other brain regions responsible for emotional and mood regulation, such as the amygdala and prefrontal cortex. Any malfunction within these networks might hamper emotional processing, thus amplifying the mood-related symptoms synonymous with depression. Iron potentially holds essential significance in the survival of hippocampus neurons, thereby influencing the progression of depression.

Our findings also reveal several insights into the relationships between brain iron accumulation and AD. We identified five genes

associated with the brain iron levels of the accumbens nucleus: *SHANK1*, *CST9L*, *HIGD1C* and *SLC39A8*. Notably, the cognitive deficits typical of AD often coincide with synaptic loss attributed to disruptions in the postsynaptic density. Such disruptions are evidenced by a marked reduction in *SHANK1* protein levels[36]. Given the accumbens nucleus's role in dopamine regulation[37], the research highlighted the potential consequence of perturbed dopaminergic signaling in the context of AD[38]. Furthermore, excess iron levels in neural tissues can induce oxidative stress, known to adversely affect neural integrity and functionality[39]. Such oxidative stress has been linked to disturbing neurotransmission, particularly of dopamine, which in turn influences

the function of the accumbens nucleus[40]. In PheWAS, we also find that *SLC39A8* is associated with the cognition test scores (i.e., fluid intelligence and prospective memory), which aligns with the known characteristics of AD as a progressive neurodegenerative disorder characterized by memory loss and cognitive deficits.

A major strength of our research lies in its pioneering use of the most extensive WES analyses for brain iron accumulation to date. This approach has led us to identify multiple genes not previously reported, further enriching our understanding of the genetic architecture behind brain iron accumulation. The current study also has some potential limitations. Our analysis predominantly encompasses participants of European genetic ancestries and is constrained by the available sample size for brain imaging data. Incorporating cortical brain regions in our study might have enhanced our findings, offering a more comprehensive understanding of the genetic structure and associations concerning regional brain iron levels. In addition, a large external validation dataset would be beneficial to verify the solidity of our findings.

In conclusion, our investigation identified 36 genes associated with brain iron accumulation across multiple brain regions. Many of these genes are enriched in pathways related to iron transport and homeostasis and are linked to iron-related brain disorders. Furthermore, our study also revealed several causal pathways from regional brain iron accumulation to disorders such as PD, AD and depression. These findings provided insights into the genetic architecture of brain iron accumulation and uncovered the important relationships of brain iron with several brain disorders and behavioral traits. We anticipate that our findings will serve as a groundwork for future research, aiding in the elucidation of how these genes impact brain iron levels and contribute to the onset and progression of brain disorders.

## Methods

### Study population
The UK Biobank (UKB) (https://www.ukbiobank.ac.uk/) is a population-based prospective cohort of ~500,000 participants aged from 40 to 69 years old at enrollment between 2006 and 2010[41], among which whole-exome sequencing data are available for 454,787 participants. UKB received ethical approval from the National Health Service National Research Ethics Service (reference: 11/NW/0382) and all participants provided written informed consent. This study was conducted under application number 19542.

### Brain imaging-derived phenotypes
We used an automated quantitative susceptibility mapping (QSM) pipeline[8], based on the susceptibility-weighted MRI (swMRI) data, to measure the brain iron. We used swMRI data from 37,213 subjects in the UK Biobank (release 2022). The detailed MRI data acquisition, quality control, and QSM processing pipelines have been reported in the original paper[8]. The individual space voxel-wise QSM maps are downloaded and warped to the MNI152 standard space based on the warp field maps. We extracted the median QSM values from 26 subcortical and cerebellum structures as shown in Fig. 1. Among them, 16 of the 26 subcortical structures were defined based on the original paper (the accumbens, amygdala, caudate, hippocampus, pallidum, putamen, substantia nigra and thalamus, both left and right)[8]. The corresponding field IDs for these imaging-derived phenotypes are listed in Supplementary Data 20. We further selected 10 additional regions of interest (the red nucleus, subthalamic nucleus, globus pallidus externa, globus pallidus interna and dentate, both left and right), based on the segmentation masks of the multi-contrast PD25 atlas[42] and the deep cerebellar nuclei probabilistic atlas[43,44].

### Exome sequencing and quality control
Whole-exome sequencing was conducted on 454,787 participants in the Regeneron Genetics Center (RGC) and protocols were described in

detail elsewhere[10]. The OQFE WES pVCF files in GRCh38 human reference genome build[45] were utilized in this study, and we performed additional quality control similar to the previous study[46]. First, multi-allelic sites were split into bi-allelic sites and all calls that have a low genotype quality and extremely low/high genotype depth were set to no-call. Variants with call rate <90%, Hardy–Weinberg $P$-value $< 10^{-15}$, and in Ensembl low-complexity regions were excluded. Samples withdrawn from the study, duplicates, with discordance between self-reported and genetically inferred sex, samples whose Ti/Tv, Het/Hom, SNV/indel, and number of singletons exceed 8 standard deviations from the mean were removed. We used King software to calculate the kinship coefficient using the high-quality variants (MAF > 0.1%, missingness <1%, HWE $P > 10^{-6}$ and two rounds of pruning using --indep-pairwise 200 100 0.1 and --indep-pairwise 200 100 0.05). Unrelated samples were defined using the kinship coefficient threshold at 0.0884, indicating the 2nd relatedness. To maximize the sample size, participants related to multiple other individuals were first iteratively removed until none remained. Then, one of the remaining kinship pairs was removed at random. In this study, samples were mainly restricted to White British (filed 22006) and ancestry-specific principal components were calculated, which were used in the following analysis.

### Variant annotation
SnpEff[47] was used to annotate rare variants (MAF < 1%), and for those annotated with multiple consequences, the most severe consequence was kept for each gene transcript. Loss of function (LOF) was defined for variants annotated as stop gained, start lost, splice donor, splice acceptor, stop lost, or frameshift. Likely deleterious missense was determined if variants were consistently predicted as deleteriousness in SIFT[48]; PolyPhen2 HDIV and PolyPhen2 HVAR[49]; LRT[50]; and MutationTaster[51]. As for common variants (MAF ≥ 1%), ANNOVAR[52] was utilized to annotate variants using refGene as a reference panel, and those annotated as exonic, UTR3, or UTR5 were kept in the following single-variant EWAS analysis.

### Data partition
In this study, we partitioned the entire dataset into two subsets: a discovery set and a replication set. This partitioning was based on the availability of longitudinal swMRI scans for individual subjects. Two independent association tests were conducted as follows: Discovery set: This subset comprised baseline imaging data from 26,789 subjects who did not have longitudinal swMRI scans available. Statistical significance in the discovery set was determined using the genome-wide significance threshold. Replication set: This subset consisted of repeated scan brain imaging data from 3039 subjects. Statistical significance in the replication set was set at an uncorrected $p$-value threshold of <0.05, following the approach utilized in a previous study[53].

### Exome-wide association analysis
EWAS was performed in the discovery and replication sets using unrelated British with both whole exome sequence (WES) and swMRI brain imaging data available using a generalized mixed model implemented in SAIGE-GENE+[54] A total of 18,800 variants were analyzed. Rare pLOF and likely deleterious missense variants with MAF < 0.01 were collapsed in each gene, and the EWAS was calculated using the Burden test, SKAT, and SKAT-O, adjusting for age, sex, and the first ten principal components. We constructed six frequency-function collapsing masks for each gene in the gene-based collapsing test: for the frequency of variants, including MAF < 0.01, <0.001, and <0.0001; for the function of variants, pLOF and pLOF+ likely deleterious missense variants. For common variants, a single-variant association analysis was performed, adjusting for the same covariates as in the gene-based collapsing test. A sparse genetic relationship matrix was constructed using the high-quality variants with the recommended relative coefficient cutoff of 0.05. Bonferroni correction was applied and the significance threshold was set to

$p < 1.7 \times 10^{-8}$. This is computed as $0.05/(18,800 \times 3 \times 2 \times 26)$, where 18,800 is the number of candidate rare genes, 3 is the number of minor allele frequency cutoffs, 2 is the number of variants annotation groups, and 26 is the number of brain regions.

## Robustness of EWAS

To assess the robustness of the results and detect the variants driving the association in the collapsing test, we further performed leave-one-variant-out analysis. The variant maximized the $p$-value after removing the collapsing test was identified as the driving variant. For the significant gene–phenotype associations found in the gene-based EWAS, conditional analyses were further performed to evaluate the influence of the nearby common loci (defined as independent index variants after clumping (--clump-p1 $1 \times 10^{-5}$ --clump-r2 0.01), ±500 kb of the identified gene, MAF > 0.5% in the UKB imputed data). For the significant gene–phenotype associations, gene-based collapsing tests were reperformed to condition on nearby common variants signals. For single variant EWAS, independent significant SNPs were further identified ($p < 1.0 \times 10^{-5}$ and r2 ≤ 0.6 within a 1 Mb window). Independent significant SNPs were then clumped to obtain lead SNPs ($r^2 \le 0.1$ within a 1 Mb window). Genomic loci were defined by merging lead SNPs within 250 kb. In addition, we performed subgroup analysis based on sex with the same covariates adjusted except for the subgroup factor.

## Common variants association analysis

For common exonic variants (MAF 1%), a single variant association analysis was conducted among the unrelated Caucasian cohort using SAIGE-GENE+ [54], adjusting age, gender, and ten principal components. Lead SNPs were identified as independent significant SNPs which meet significant thresholds and are independent of other significant SNPs with $r^2 < 0.1$ within a 1 Mb window. The significance threshold was set to $4.60 \times 10^{-8}$ ($0.05/(41,790 \times 26)$, Bonferroni correction for 41,790 exonic coding SNPs in 26 brain regions).

## Tissue and pathway enrichment analysis

Tissue enrichment analysis was performed by the GENE2FUNC function in FUMA [55] with all mapped genes in EWAS as input. Briefly, Differentially Expressed Gene (DEG) sets were pre-calculated by performing a two-sided $t$-test for any one type of tissue against all other tissues of 54 tissue types based on data from the GTEx database [56]. Then hypergeometric tests were used to test brain iron-associated genes against each of the DEG sets. For pathway enrichment analysis, hypergeometric tests were also performed to test if brain iron-associated genes are overrepresented in any of the pre-defined gene sets, covering Gene Ontology, Reactome, GWAScatalog, and Immunologic signatures.

## Single-nucleus RNA sequencing data source and analyses

We used single-nucleus RNA sequencing (snRNA-seq) data of human brain vasculature obtained from a recent study conducted by Garcia et al. on the Gene Expression Omnibus database with the accession ID: GSE173731 [57]. The clustering and annotation of the cell types were conducted via the metadata file provided by the authors. We also computed the gene set score of the brain iron-associated genes using results from both single and gene-based EWAS, using *AddModuleScore* function. The primary analysis and subsequent visualization were conducted using the R package Seurat.

## Protein–protein interaction network

Protein–protein interaction based on the significant gene set derived from the single variant and gene-based EWAS results were investigated using the human STRING database [58]. Interactions with a confidence score of at least medium confidence were extracted and subsequently visualized in Cytoscape [59] (version 3.9.0). Proteins were further clustered using the Markov clustering (MCL) algorithm to investigate the functional clusters using the default settings [60].

## Phenome-wide association study

To explore broader phenome-wide associations and underlining mechanisms for the brain iron-related genes or variants derived from single and gene-based EWAS results, we investigated their associations with additional phenotypes. We mainly focused on brain structures ($N = 220$: thickness, surface area, volume for 68 cortical regions and volume for 16 subcortical structures), biochemistry ($N = 30$), inflammatory ($N = 11$) markers, metabolomics ($N = 249$), and proteomics ($N = 1463$). As for rare variants, gene-based linear mixed models in SAIGE-GENE+ were employed. And for common variants, single-variant association analysis was performed using linear regression by PLINK v2. Both models were adjusted for age, sex, and the first ten genetic principal components.

Standard Siemens Skyra 3T running VD13A SP4 with a 32-channel head coil was used to acquire the T1-weighted neuroimaging data with a resolution of $1 \times 1 \times 1$ mm (Field 20252) (detailed acquisition protocol can be found at https://biobank.ndph.ox.ac.uk/showcase/showcase/docs/brain_mri.pdf). The cortical surface areas, volumes, and mean thickness for 68 cortical regions were extracted based on FreeSurfer's surface templates using aparc atlas [61].the volume for 16 subcortical regions was estimated via FreeSurfer's aseg tool [62].

Biochemistry or inflammatory markers were obtained from blood count data (Category 100081) and blood biochemistry data (Category 17518) based on UK Biobank blood samples (detailed protocol can be found at https://biobank.ndph.ox.ac.uk/showcase/label.cgi?id=100080). Four blood cell count ratios were additionally calculated for downstream analysis, including the neutrophils to lymphocytes ratio (NLR), platelet-to-lymphocyte ratio (PLR), lymphocyte-to-monocyte ratio (LMR), and the systemic immune-inflammation index (SII).

Nuclear magnetic resonance (NMR) metabolomics data (Category 220) were acquired from randomly selected EDTA plasma samples using a high-throughput NMR-based metabolic biomarker profiling platform. This platform covers 249 metabolic spanning multiple metabolic pathways, including lipoprotein lipids, fatty acids, fatty acid compositions, and various low-molecular-weight metabolites (detailed protocol can be found at https://biobank.ndph.ox.ac.uk/showcase/label.cgi?id=220).

Proteomics data (Category 1838) of 1463 proteins in plasma were measured by Olink Explore platform, using Proximity Extension Assay (PEA) (detailed information on sample collection, processing, normalization, and quality control procedures can be found at https://biobank.ndph.ox.ac.uk/showcase/label.cgi?id=1839).

## Causal relationship between regional brain iron and brain disorders

To investigate the causal relationships between QSM and multiple brain disorders, we first performed a Genome-Wide Association Study (GWAS) using imputed SNP (to the Haplotype Reference Consortium) genotype data obtained from the UK Biobank resource [63]. A total of 8,445,740 SNPs were included in the GWAS. Samples were mainly restricted to white British and those who were used in computing the principal components. Individuals with a missing genotype rate > 0.05, with mismatch self-reported (Data field 31) and genetic sex (Data field 22001), with abnormal sex chromosome aneuploidy, and have more than 10 putative third-degree relatives been further removed. We also excluded variants with call rate < 0.95, MAF < 0.01, Hardy–Weinberg $P$-value < $10^{-6}$, or imputation quality score < 0.5. GWAS were independently performed for each phenotype using linear regression models implemented in PLINK2 [64]. Covariates included age, sex, and the first ten genetic principal components. A total of 26,776 to 28,129

participants with phenotype and covariates available for 26 subcortical and cerebellum structures were included in the final GWAS.

Our primary two-sample MR analyses were conducted using regional brain iron as exposure, and diseases as outcomes. For diseases, we leveraged GWAS summary data of iron-related brain disorders, including Parkinson's disease (Ncase = 33,674, Ncontrol = 449,056)[65], Depression (excluding 23andMe and UK Biobank: Ncase = 45,591, Ncontrol = 97,674)[66], Bipolar (Ncase = 20,352, Ncontrol = 31,358)[67], and Alzheimer's disease (Ncase = 21,982, Ncontrol = 41,944)[68]. Instrumental variables (IV) were selected based on a significant level ($p < 5 \times 10^{-8}$) and followed by LD clumping ($R^2 > 0.001$). Then IV from the exposure and outcome data were harmonized to the same effect alleles. $F$-statistics were computed to assess the strength of the instruments. When only a single SNP was available, the Wald ratio was used to estimate the causality of exposure to outcome. When more than one SNP was available, the inverse-variance weighted (IVW) with multiplicative random effects method was employed[69]. MR-PRESSO test[70] was used to detect outliers, and if an outlier was detected, the original p-value was replaced by the outlier-corrected $p$-value. The $q$-value FDR approach was used to correct for multiple comparisons across brain regions and diseases[71].

To assess the robustness of our analysis, several sensitivity analyses were conducted. This involved using different MR methods, including MR Egger[72], Wald ratio[73], and Weighted median[74]. The intercept of MR Egger was used to identify the presence of directional pleiotropy. Considering the similarity of genetic architecture between different brain regions, LASSO feature selection (mv_lasso_feature_-selection () function in "TwoSampleMR") followed by a Multivariable MR (MVMR) was performed to further assess whether the causal effects were independent[75]. In addition, we also performed a reverse MR (from neurologic and psychiatric disorders to regional brain iron) to infer the direction of causality. The MR analysis was performed using the "TwoSampleMR" version 0.5.6 in R version 4.2.

### Power analysis

We simulated 1000 datasets for each combination of the estimated effect sizes (i.e., based on the regression coefficient and its estimated errors) and the cMAC per gene of our analysis, based on the method in ref. 11. Carrier status was randomized across $N = 29,828$ (sample size for our collapsing tests) participants. A linear regression model was used to test for the association with a threshold of $p < 1.7 \times 10^{-8}$ (corresponding to the significance threshold in our collapsing tests).

### Statistics and reproducibility

The code used in the paper is made publicly available for reproducibility purposes. Statistical analyses are given as well. There is no randomness for all results presented in this study.

### Reporting summary

Further information on research design is available in the Nature Portfolio Reporting Summary linked to this article.

## Data availability

The main data, including the individual-level phenotypic and genetic data used in this study, were accessed from the UK Biobank under application number 19542 and were available through UKB. The EWAS summary statistics are available at https://doi.org/10.5281/zenodo.11170064[76].

## Code availability

All software and R packages used to perform the analyses in this work are freely available online: SnpEff, https://pcingola.github.io/SnpEff/; SAIGE-GENE+, https://github.com/saigegit/SAIGE; R, https://www.r-project.org; PLINK2, https://www.cog-genomics.org/plink/2.0/; FUMA, https://fuma.ctglab.nl; MAGMA, https://ctg.cncr.nl/software/magma/; STRING, https://www.string-db.org/. The scripts[76] used to conduct the main analyses are available at https://github.com/weikanggong/BrainIronWES.

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

## Acknowledgements

We gratefully thank all UK Biobank participants for their time and UK Biobank team members for collating the data. J.-T. Yu was supported by grants from the Science and Technology Innovation 2030 Major Projects (2022ZD0211600), National Natural Science Foundation of China (82071201, 81971032, 92249305), Shanghai Municipal Science and Tech-nology Major Project (No.2018SHZDZX01), Research Start-up Fund of Huashan Hospital (2022QD002), Excellence 2025 Talent Cultivation Pro-gram at Fudan University (3030277001), Shanghai Talent Development Funding for The Project (2019074), and ZHANGJIANG LAB, Tianqiao and Chrissy Chen Institute, and the State Key Laboratory of Neurobiology and Frontiers Center for Brain Science of Ministry of Education, Fudan Uni-versity. W. Cheng was supported by grants from the National Natural Sciences Foundation of China (no. 82071997) and the Shanghai Rising-Star Program (no. 21QA1408700). J.F. Feng was supported by the National Key R&D Program of China (No.2018YFC1312904 and No. 2019YFA0709502), the Shanghai Municipal Science and Technology Major Project (No. 2018SHZDZX01), the 111 Project (No. B18015), Shanghai Center for Brain Science and Brain-Inspired Technology and Zhangjiang Lab. Figure 1 was partly generated using Servier Medical Art, provided by Servier, licensed under a Creative Commons Attribution 4.0 unported license.

## Author contributions

W.K. Gong, W. Cheng, J.-T. Yu and J.F. Feng designed the study. W.K. Gong, Y. Fu, B.-S. Wu, and J.N. Du conducted the main analyses and drafted the manuscript. W.K. Gong, Y. Fu, B.-S. Wu, J.N. Du, L. Yang, Y.-R. Zhang, S.-D. Chen, J.J. Kang contributed to data collection and analyses. W.K. Gong, J.-T. Yu, W. Cheng, Y. Mao, Q. Dong, L. Tan, and J.F. Feng critically revised the manuscript. All authors reviewed and approved the final version.

## Competing interests

The authors declare no competing interests.
