## [Peer Review File · Nature Communications]

Whole-Exome Sequencing Identifies Protein-Coding Variants Associated with Brain Iron in 29,828 IndividualsEditorial Note: This manuscript has been previously reviewed at another journal that is not operating a transparent peer review scheme. This document only contains reviewer comments and rebuttal letters for versions considered at *Nature Communications* .

REVIEWERS' COMMENTS

Reviewer #1 (Remarks to the Author):

The authors have now adequately addressed my concerns.

Reviewer #1 (Remarks on code availability):

Code is a series of shell scripts, README file doesn't contain instructions only a general description (i.e. code for this article). I didn't run the scripts, but they seem to be straightforward.

Response to the reviewers

The authors sincerely appreciate the critical reviews of the paper. We have now revised the paper to carefully address all the points raised. Our responses below are preceded by "--", and changes made to the paper are shown below within "...", and in red font in the revised paper.

Reviewer #1:

Remarks to the Author:

Code is a series of shell scripts, README file doesn't contain instructions only a general description (i.e. code for this article). I didn't run the scripts, but they seem to be straightforward.

--We updated the code, and provided a detailed README in the Github, as follows:

The code used in the paper Gong et al. "Whole-Exome Sequencing Identifies Protein-Coding Variants Associated with Brain Iron in 29,828 Individuals". The data utilized for these analyses originates from the UK Biobank cohort.

For the operation of this project, the installation of the following software or package is required: SAIGE-GENE+ R package (<https://github.com/saigegit/SAIGE>) TwoSampleMR version 0.5.6 R package ([MRCIEU/TwoSampleMR: R package for performing 2-sample MR using MR-Base database \(github.com\)](https://github.com/MRCIEU/TwoSampleMR)) PLINK v2.0 (<https://www.cog-genomics.org/plink/2.0/>)

The scripts EWAS_code.sh, Common_code.sh, and LOVO_code.sh utilize the SAIGE-GENE+ package for the identification of genetic variants associated with brain iron accumulation. Specifically, EWAS_code.sh is used for detecting rare variants, Common_code.sh for common variants, and LOVO_code.sh is employed to identify the most significant variants driving the observed associations.

The scripts conditional_code.sh utilize the SAIGE-GENE+ package and PLINK software to assess the interaction of rare and common variants.

The scripts MR_code_example.R utilize the TwoSampleMR package to validate 490 the causal relationship and the simulate_WES_power.R was used to show the power calculation of the cohorts.